# Gut microbial composition is altered in sarcopenia: A systematic review and meta-analysis of clinical studies

**Xiaohong Mai[1], Shuyi Yang[2], Qifeng Chen[3], Kangkang Chen**[3]*

**1** Department of Geriatric Psychiatry, Shaoxing Seventh People's Hospital, Shaoxing, China, **2** Department of Microbial Detection, Shaoxing Center for Disease Control and Prevention, Shaoxing, China, **3** Department of Non-Communicable Diseases Control and Prevention, Shaoxing Center for Disease Control and Prevention, Shaoxing, China

\* sxcdc_ck@163.com

**Data Availability Statement:** Other researchers can find all data underlying the findings described in our manuscript within our manuscript itself.

**Funding:** The author(s) received no specific funding for this work.

## Abstract

Increasing evidence has shown that gut microbiota (GM) was involved in the pathophysiology of musculoskeletal disorders through multiple pathways such as protein anabolism, chronic inflammation and immunity, and imbalanced metabolism. We performed a systematic review and meta-analysis of human studies to evaluate GM diversity differences between individuals with and without sarcopenia, and explore bacteria with potential to become biomarkers. PubMed, Embase and Cochrane library were systematically searched from inception to February 16, 2024. Studies were included if they (1) sampled adults with sarcopenia, and (2) performed GM analysis and reported α-diversity, β-diversity or relative abundance. The methodological quality of included studies and the certainty of evidence were assessed through the Joanna Briggs Institute critical appraisal checklist for analytical cross-sectional studies and the Grades of Recommendation, Assessment, Development and Evaluation (GRADE) Working Group system, respectively. Weighted standardized mean differences (SMDs) and corresponding 95% confidence intervals (CIs) were estimated for α-diversity indices using a fixed-effects and a random-effects model. Beta diversity and the relative abundance of GM were summarized qualitatively. A total of 19 studies involving 6,565 participants were included in this study. Compared with controls, significantly moderate decrease in microbial richness in participants with sarcopenia were found (Chao1: SMD = -0.44; 95%CI, -0.64 to -0.23, $I^2$ = 57.23%, 13 studies; observed species: SMD = -0.68; 95%CI, -1.00 to -0.37, $I^2$ = 66.07%, 5 studies; ACE index: SMD = -0.30; 95% CI, -0.56 to -0.04, $I^2$ = 8.12%, 4 studies), with very low certainty of evidence. Differences in β-diversity were consistently observed in 84.6% of studies and 97.3% of participants. The detailed analysis of the gut microbial differential abundance identified a loss of *Prevotellaceae*, *Prevotella*, and *Megamonas* in sarcopenia compared with non-sarcopenia. In conclusion, sarcopenia was found to be associated with reduced richness of GM, and supplementing intestinal bacteria described above may contribute to preventing and treating this muscle disease. The research protocol was registered and approved in PROSPERO (CRD42023412849).

**Competing interests:** The authors have declared that no competing interests exist.

**Abbreviations:** GM, gut microbiota; SMD, standardized mean difference; CI, confidence interval; PRISMA, Preferred Reporting Items for Systematic Reviews and Meta-Analyses; BMI, body mass index; JBI, The Joanna Briggs Institute; SD, standard deviation; HF, heart failure; AWGS2019, the Asian Working Group for Sarcopenia 2019; EWGSOP, the European Working Group on Sarcopenia in Older People; FNIH, the Foundation for the National Institutes of Health; SCFAs, short-chain fatty acids; mTOR, mammalian target of rapamycin; IGF-1, insulin-like growth factor-1; RCT, randomized controlled trial; GRADE, The Grades of Recommendation, Assessment, Development and Evaluation.

## Introduction

Sarcopenia is a progressive and generalized skeletal muscle disorder that is characterized by loss of muscle mass, low muscle strength, and reduced physical performance [1, 2]. Individuals with sarcopenia are commonly at a higher risk of adverse health events, including falls, chronic disease states, and even all-cause mortality [3–5]. Since early sarcopenia is asymptomatic and there has been a lack of effective ways to treat it [6], the burden of sarcopenia is high, and continues to increase. It is reported that the prevalence of sarcopenia is expected to increase from 9% in 2019 to almost 23% by 2100 [7]. It is therefore necessary to explore new avenues for taking effective precautions, improving screening accuracy, and providing treatment, wherein increasing evidence has showed that gut microbiota (GM) composition may achieve these goals.

The human GM is composed of 10–100 trillion microorganisms which play a significant role in muscle health. The GM is deeply involved in metabolic interactions like food decomposition and nutrient intake [8, 9]; also, the by-products of this biological process such as tryptophan and short-chain fatty acids (SCFAs) could promote the myofibril synthesis [10, 11]. Additionally, the GM was found to be associated with systemic chronic inflammation and host's immune responses. The normal GM could enhance intestinal barrier to balance pro- and anti-inflammatory cytokines [12]; conversely, the disordered GM may have led to an increased level of lipopolysaccharide, indoxyl sulfate, and trimethylamine-N-oxide, which induced a pro-inflammatory status [13–15]. More importantly, people with advanced age or inactivity were more vulnerable to the GM dysbiosis, with lower abundance of beneficial microbes and higher harmful bacterial metabolites [16, 17]. In epidemiology studies, malnutrition, disease, aging, and inactivity were considered four main factors for developing and worsening sarcopenia [2]; therefore the concept of the 'gut–muscle axis' has been raised to study the direct and indirect relationships.

The advances in high-throughput sequencing technologies make it possible to study gut ecosystem, which provides new insight into the 'gut–muscle axis' theory. High diversity of GM generally means better body health, and alpha diversity is a comprehensive indicator to measure it, including richness (number of species) and evenness (how well each species is represented) [18]. As a promising preventive and therapeutic target, the number of clinical studies on associations between the GM composition and sarcopenia have burgeoned since 2021 worldwide. However, the findings derived from these human studies were not completely consistent. For example, Ni Lochlainn *et al.* [19] showed that no significant difference was found between individuals with and without sarcopenia in terms of alpha diversity; conversely, Han *et al.* [20] reported that alpha diversity was significant reduced in sarcopenic individuals. So far, no meta-analysis of human studies has been conducted to evaluate GM alterations in sarcopenia. This study was therefore designed to fill this gap, and to further explore the predominant bacteria which may serve as biomarkers.

## Materials and methods

This meta-analysis was performed according to the PRISMA statement (Preferred Reporting Items for Systematic Reviews and Meta-Analyses) [21]. The PRISMA2020 checklist is shown in S1 Table. The research protocol was registered and approved in PROSPERO (CRD42023412849).

### Data sources

PubMed, Embase and Cochrane library were retrieved from inception to February 16, 2024 by using (i) the MeSH term "Sarcopenia" and (ii) a cluster of text words on gut microbiota to

identify published studies comparing the composition of the gut microbiota in adults with and without sarcopenia. The detailed search strategies are shown in S2 Table. Additionally, reference lists of the included studies were retrieved for any relevant studies. Only English publications were considered.

## Selection criteria

Studies were included if they (1) sampled adults with sarcopenia, and (2) performed gut microbiota analysis and reported α-diversity, β-diversity or relative abundance. Exclusion criteria were as follows: (1) without a control group; (2) duplicate studies (only the most detailed study was included in the analysis); and (3) editorials, conference proceedings, abstracts or case reports. Two researchers (X.M. and S.Y.) independently screened the titles and abstracts to evaluate the potential studies. If a study was relevant, the full article was obtained for further evaluated. Disagreements were determined by discussion or with a third researcher (Q.C.).

## Data extraction

Two independent researchers (X.M. and S.Y.) extracted the following data: lead author, publication year, country, type of participants, sample size, mean age, female ratio, body mass index (BMI), definition of sarcopenia, method of gut microbiota assessment, and data on outcomes (α-diversity, β-diversity and relative abundance). The extracted data were checked for accuracy by a third researcher (Q.C.).

## Risk of bias assessments and certainty assessments

The Joanna Briggs Institute (JBI) critical appraisal checklist for analytical cross-sectional studies was used to assess the methodological quality of included studies. Two researchers (X.M. and S.Y.) performed the rating independently by examining eight items: 1) "Were the criteria for inclusion in the sample clearly defined?", 2) "Were the study subjects and the setting described in detail?", 3) "Was the exposure measured in a valid and reliable way?", 4) "Were objective, standard criteria used for measurement of the condition?", 5) "Were confounding factors identified?", 6) "Were the confounding factors identified?", 7) "Were the outcomes measured in a valid and reliable way?", and 8) "Was appropriate statistical analysis used?". Each item is rated "Yes", "No" or "unclear". Studies that scored five or more "Yes" were regarded as high quality and included in the meta-analysis. Any disagreements were resolved in a consensus meeting with a third researcher (Q.C.) as a referee.

The Grades of Recommendation, Assessment, Development and Evaluation (GRADE) Working Group system were used to assess the certainty of the body of evidence associated with outcomes [22]. The quality of evidence resulted from observational studies began as low certainty evidence (score = -2), and could be rated down (-1 or -2) for serious or very serious concerns on the basis of following domains: risk of bias, inconsistency, imprecision, indirectness, publication bias, and could be rated up (+1 or +2) on the basis of upgrading domains including large effect, dose-response, opposing plausible residual confounding and bias. The final scores of $\geq 0$, -1, -2, $\leq$ -3 were defined as high certainty, moderate certainty, low certainty, and very low certainty, respectively.

## Statistical analysis

Meta-analyses were performed to assess the differences in alpha diversity between sarcopenia and non-sarcopenia groups. In this part, sample size, mean and standard deviation (SD) were collected for analysis. If the required data were presented by median and interquartile range,

data was conducted through two ways: 1) a web-based tool (https://www.math.hkbu.edu.hk/~tongt/papers/median2mean.html) if the data are not significantly skewed; or 2) an alternative validated formula listed by Wan *et al.* [23] if the data are significantly skewed. Where necessary, WebPlotDigitizer 4.6 (https://apps.automeris.io/wpd/index.zh_CN.html) was used to extracted numerical data from figures. Heterogeneity between summary data was assessed using the $I^2$ statistic. $I^2$ <50% reflected mild to moderate heterogeneity, and >50% severe heterogeneity. A random effects model was used to calculate the weighted standardized mean difference (SMD) and 95% confidence intervals (CIs) unless mild to moderate heterogeneity was detected, then a fixed effects model was used. Effect size was categorized as small (SMD <0.25), moderate (SMD = 0.25–0.75), or large (SMD >0.75). To ascertain robustness of findings, sensitivity analyses were performed by repeating with the random-effect method for severe heterogeneity. To identify predictors and explore sources of heterogeneity, exploratory sub-analyses were conducted based on variables, including clinical characteristics (region, definition of sarcopenia, research setting, age, female ratio, BMI and Method to measure gut microbiota) and study characteristics (publication year and sample size). For variables without appropriate threshold to categorize patients, the medians were calculated according to values reported in each study. Publication bias was estimated using funnel plots and Egger's regression intercept analysis if the number of studies included were ≥10. Analyses were performed with Stata version 16 (Stata Corp., College Station, TX, USA).

Qualitative reviews were performed to pool the current evidence regarding beta diversity and the relative abundance of gut microbes due to the limited overlap. Given high likelihood of false-positive results reported before [24], microbial taxa reported in only one study were excluded. If a microbial taxon was reported in both two studies and altered in a consistent direction, we labeled it increased or decreased, which means the necessity of further validation. If a microbial taxon was reported in 3 or more studies, we labeled it increased, decreased, or not consistent based on proportion greater than 70%, which means a potential relationship with sarcopenia.

All tests were 2-tailed, and $P < 0.05$ was considered statistically significant.

## Results

### Studies retrieved and characteristics

A total of 338 publications were retrieved through the initial literature search. After removing duplicates and non-relevant abstracts, 28 full texts were further assessed for eligibility. Of these, 9 studies were excluded: 3 did not provide data for alpha diversity, beta diversity or relative abundance; 2 did not compare gut microbiota diversity in individuals with or without sarcopenia; 2 were conference abstracts; 1 was a review; 1 was not related to gut microbiota. Finally, a total of 19 studies involving 998 adults with sarcopenia and 5,567 controls were included for the final analysis (Fig 1).

Among the 19 studies, 10 recruited community dwellers [19, 20, 25–32], 4 recruited patients who were hospitalized [33–36], 2 recruited patients with cirrhosis [37, 38], 1 recruited hemodialysis patients [39], 1 recruited patients with heart failure (HF) [40], and 1 recruited patients with chronic kidney disease [41]. Most studies (14 [73.7%]) were conducted in Asia (China and Korea) [20, 27–36, 38–40], 5 (26.3%) in Europe (Italy and United Kingdom) [19, 25, 26, 37, 41]. Generally, the Asian Working Group for Sarcopenia 2019 (AWGS2019) was used to define sarcopenia in studies conducted in Asia, and the European Working Group on Sarcopenia in Older People (EWGSOP) or the Foundation for the National Institutes of Health (FNIH) in Europe. Most studies (15 [78.9%]) were conducted in older adults ≥60 years [19, 20, 25–28, 30, 31, 33–37, 40, 41], whereas 4 (20.0%) studies in adults <60 years [29, 32, 38, 39].

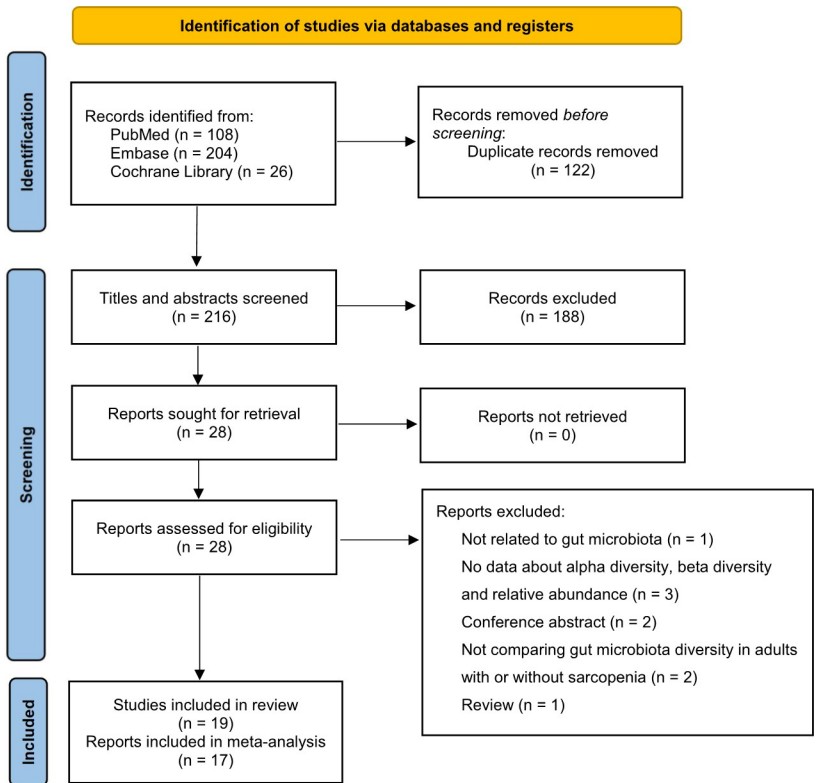

**Fig 1. Literature search and screening process.**

The mean age ranged from 45.9 to 83.1 years. With regard to methodology of gut microbiota assessment, 15 (78.9%) studies used 16S ribosomal RNA sequencing, followed by 2 studies (10.5%) using shotgun metagenomics. The detailed characteristics of the studies and patients are given in Table 1. According to the JBI critical appraisal checklist, all studies included in the review were high in terms of the methodological quality (S3 Table). No significant publication bias was observed for richness index ($P$ = 0.236; S1 Fig).

## Alpha diversity

A total of 7 indices were used to assess alpha diversity, including richness (Chao1, observed species, ACE index and index of species richness), as well as diversity (Shannon, Simpson and Faith phylogenetic diversity). According to the findings of the GRADE analysis, the certainty of evidence for these indices were very low (S4 Table).

Thirteen studies reported richness as an outcome [20, 25–27, 31–37, 39, 40]. The pooled estimate showed a significant decrease in sarcopenic group with a moderate effect size (SMD = -0.47; 95% CI: -0.62 to -0.32) and severe heterogeneity ($I^2$ = 57.85%) (Fig 2). Chao1, observed species, ACE index and index of species richness were reported in twelve [20, 26, 27, 31–37, 39, 40], five [20, 33–35, 40], four [31, 32, 36, 39], and one [25] studies, respectively. The pooled SMDs (95% CIs) were -0.44 (-0.64 to -0.23) for Chao1, -0.68 (-1.00 to -0.37) for observed species, -0.30 (-0.56 to -0.04) for ACE index and -0.13 (-1.13 to 0.86) for index of species richness. There were no significant differences between these indices ($P$ = 0.29). In sensitivity analysis, the fixed-effect method did not change the result, suggesting robustness of analysis to the data-effect model.

**Table 1. Characteristics of the included trials and participants.**

| Study | Country | Participants | Sample size (n) | Mean age | Female ratio (%) | Mean BMI | Definition of sarcopenia | Method of gut microbiota assessment | Outcome | Alpha diversity (mean±sd) |
|---|---|---|---|---|---|---|---|---|---|---|
| Picca 2019 | Italy | Community dwellers | S: 18 NS: 17 | S: 75.5 NS: 73.9 | S: 56.0 NS:29.0 | S: 32.1 NS: 26.3 | FNIH | 16S rRNA sequencing of V3-V4 | Alpha diversity, relative abundance | Chao1(S:549.6 ±161.4; NS:598.1 ±126.3) |
| Ticinesi 2020 | Italy | Community dwellers | S: 5 NS: 12 | S: 77.0 NS: 71.5 | S: 80.0 NS: 83.3 | S: 24.3 NS: 27.4 | EWGSOP1 | Shotgun metagenomic sequencing | Alpha diversity, beta diversity, relative abundance | Index of species richness (S:78.0 ±28.0; NS:81.0±18.0) |
| Kang 2021 | China | Hospitalized | S: 11 NS: 60 | S: 76.5 NS: 68.4 | S: 63.6 NS:55.0 | S: 20.7 NS: 23.7 | AWGS2019 | 16S rRNA sequencing of V3-V4 | Alpha diversity, beta diversity, relative abundance | Chao1(S:263.7 ±157.6; NS:635.7 ±736.7) Observed species (S:202.4±91.4; NS:418.5±411.3) |
| Margiotta 2021 | Italy | Chronic kidney disease | S: 18 NS: 45 | S: 83.1 NS: 79.7 | S: 11 NS: 38 | S: 25.5 NS: 29.3 | EWGSOP2 | 16S rRNA sequencing of V3-V4 | Relative abundance | NR |
| Ponziani 2021[a] | Italy | Cirrhosis | S: 19 NS: 31 | S: 70 NS: 66 | S: 36.8 NS: 25.8 | S: 29 NS: 27.3 | FNIH | 16S rRNA sequencing of V3-V4 | Alpha diversity, beta diversity, relative abundance | Chao1(S:376.7 ±100.1; NS:453.3 ±145.9) |
| Ponziani 2021[b] | Italy | Cirrhosis | S: 14 NS: 36 | S: 75.5 NS: 72.5 | S: 42.9 NS: 41.7 | S: 30.0 NS: 26.2 | FNIH | 16S rRNA sequencing of V3-V4 | Alpha diversity, beta diversity, relative abundance | Chao1(S:535.3 ±136.5; NS:522.9 ±183.0) |
| Lee 2022 | Korea | Community dwellers | S: 27 NS: 33 | S: 66.5 NS: 64.8 | S: 81.5 NS: 69.7 | NR | AWGS2019 | 16S rRNA sequencing of V3-V4 | Alpha diversity, beta diversity, relative abundance | Chao1(S:559.2 ±144.4; NS:545.0 ±167.3) Shannon index (S:7.0 ±0.8; NS:6.9±0.7) Simpson index (S:1.0 ±0.1; NS:1.0±0.1) |
| Zhou 2022 | China | Hemodialysis | S: 30 NS: 30 | S: 49.9 NS: 45.9 | S: 43.3 NS: 43.3 | S: 19.9 NS: 24.1 | AWGS2019 | 16S rRNA sequencing of V3-V4 | Alpha diversity, beta diversity, relative abundance | Chao1(S:294.0±29.3; NS:308.8±25.6) ACE index(S:294.6 ±31.4; NS:310.5 ±24.7) Shannon index (S:4.6 ±0.6; NS:4.7±0.8) Simpson index (S:0.9 ±0.1; NS:0.9±0.1) |
| Wang 2022 | China | Community dwellers | S: 141 NS: 1276 | S: 72.2 NS: 62.3 | S: 48.2 NS: 60.1 | S: 21.4 NS: 24.2 | AWGS2019 | Shotgun metagenomic sequencing | Alpha diversity, beta diversity, relative abundance | Shannon index (S:11.5±0.6; NS:11.4 ±0.7) |
| Wu 2022 | China | Hospitalized | S: 88 NS: 104 | S: 77 NS: 70 | S: 47 NS: 54 | S: 22.9 NS: 23.5 | EWGSOP2 | 16S rRNA sequencing of V3-V4 | Alpha diversity, relative abundance | Chao1(S:275.4 ±125.6; NS:339.0 ±136.5) Observed species (S:134.0±57.9; NS:160.0±48.8) |
| Han 2022 | China | Community dwellers | S: 24 NS: 52 | S: NR NS: 70.0 | S: NR NS: 61.5 | S: NR NS: 22.5 | IWGS | 16S rRNA sequencing of V3-V4 | Alpha diversity, beta diversity | Chao1(S:188.9±33.9; NS:251.2±76.6) Observed species (S:188.3±37.3; NS:250.8±76.2) Shannon index (S:4.0 ±0.4; NS:4.2±0.4) |

*(Continued)*

**Table 1.** (Continued)

| Study | Country | Participants | Sample size (n) | Mean age | Female ratio (%) | Mean BMI | Definition of sarcopenia | Method of gut microbiota assessment | Outcome | Alpha diversity (mean±sd) |
|---|---|---|---|---|---|---|---|---|---|---|
| Ni Lochlainn 2023 | United Kingdom | Community dwellers | S: 129 NS: 2,862 | S: 78.2 NS: 71.6 | S: 84 NS: 89 | S: 24 NS: 27 | EWGSOP2 | 16S rRNA sequencing of V3-V4 | Alpha diversity | Shannon index (S:5.2 ±0.7; NS:5.2±0.7) |
| Peng 2023 | China | Heart failure | S: 29 NS: 33 | S: 75.1 NS: 71.8 | S: 55.2 NS: 27.3 | S: 20.3 NS: 24.2 | AWGS2019 | 16S rRNA sequencing of V3-V4 | Alpha diversity, beta diversity, relative abundance | Chao1(S:807.4 ±530.6; NS:905.8 ±195.2) Observed species (S:659.2±508.4; NS:763.1±279.9) Shannon index (S:5.5 ±1.8; NS:5.7±0.9) Simpson index (S:0.9 ±0.1; NS:0.9±0.1) |
| Yang 2023 | China | Community dwellers | S: 170 NS: 706 | S: 66.0 NS: 58.0 | S: 60.6 NS: 57.5 | S: 20.8 NS: 24.1 | AWGS2019 | NR | Beta diversity, relative abundance | NR |
| Wang 2023 | China | Community dwellers | S: 50 NS: 50 | S: 68.4 NS: 68.7 | S: 100 NS: 100 | S: 22.5 NS: 24.1 | AWGS2019 | NR | Relative abundance | NR |
| Lee 2023 | China | Cirrhosis | S: 29 NS: 21 | S: 62.7 NS: 58.8 | S: 13.8 NS: 19.0 | S: 22.4 NS: 27.5 | AWGS2019 | 16S rRNA sequencing of V3-V4 | Alpha diversity, relative abundance | Phylogenetic diversity (S:7.8±2.3; NS:9.2±2.7) Shannon index (S:4.2 ±0.7; NS:4.6±1.2) |
| Yan 2023 | China | Community dwellers | S: 17 NS: 30 | NR | S: 100 NS: 100 | NR | AWGS2019 | 16S rRNA sequencing of V3-V4 | Alpha diversity, beta diversity, relative abundance | Chao1(S:207.3 ±109.8; NS:304.4 ±170.0) ACE index(S:210.5 ±117; NS:267.4 ±83.5) Shannon index (S:4.6 ±0.6; NS:4.7±0.8) Simpson index (S:0.9 ±0.1; NS:0.9±0.1 |
| Lou 2024 | China | Hospitalized | S: 108 NS: 98 | S: 72.5 NS: 71.6 | S: 38.9 NS: 37.8 | S: 20.9 NS: 21.5 | AWGS2019 | 16S rRNA sequencing of V3-V4 | Alpha diversity, relative abundance | Chao1(S:478.1±92.2; NS:722.1±336.7) Observed species (S:363.1±76.9; NS:623.1±338.5) |
| Shan 2024 | China | Hospitalized | S: 40 NS: 40 | S: 76.5 NS: 74.2 | S: 37.5 NS: 37.5 | S: 26.5 NS: 26.6 | AWGS2019 | 16S rRNA sequencing of V3-V4 | Alpha diversity, beta diversity, relative abundance | Chao1(S:256.3±98.0; NS:267.6±114.8) ACE index(S:265.4 ±115; NS:274.0 ±124.7) Shannon index (S:3.3 ±0.6; NS:3.2±0.6) Simpson index (S:0.1 ±0.1; NS:0.1±0.1 |

(*Continued*)

**Table 1.** (Continued)

| Study | Country | Participants | Sample size (n) | Mean age | Female ratio (%) | Mean BMI | Definition of sarcopenia | Method of gut microbiota assessment | Outcome | Alpha diversity (mean±sd) |
|---|---|---|---|---|---|---|---|---|---|---|
| Zhang 2024 | China | Community dwellers | S: 31 NS: 31 | S: 55.4 NS: 49.8 | S: 51.6 NS: 51.6 | S: 21.0 NS: 22.2 | AWGS2019 | 16S rRNA sequencing of V3-V4 | Alpha diversity, beta diversity, relative abundance | Chao1(S:262.0 ±105.7; NS:277.8 ±67.6) ACE index(S:259.0 ±101; NS:272.3 ±61.3) Shannon index (S:2.6 ±1.3; NS:2.4±0.5) Simpson index (S:0.8 ±0.2; NS:0.8±0.1 |

sd:standard deviation; BMI, body mass index; S: sarcopenia; NS: Non-sarcopenia; FNIH, Foundation for the National Institutes of Health sarcopenia project; 16S rRNA: 16S Ribosomal Ribonucleic Acid; V3-V4: regions of the 16S rRNA gene; EWGSOP2, European Working Group on Sarcopenia in Older People2; AWGS2019, Asian Working Group for Sarcopenia 2019 Guidelines; IWGS, the International Working Group on Sarcopenia; NR, not reporte

Ten studies reported diversity as an outcome [19, 20, 27, 28, 31, 32, 36, 38–40]. The pooled estimate demonstrated a nonsignificant difference between sarcopenic and non-sarcopenia groups (SMD = -0.01; 95% CI: -0.08 to 0.09; $I^2$ = 0.00%) (Fig 3). Shannon, Simpson and phylogenetic diversity were reported in ten [19, 20, 27, 28, 31, 32, 36, 38–40], six [27, 31, 32, 36, 39, 40], and one [38] studies, respectively. The pooled SMDs (95% CIs) were 0.02 (-0.08 to 0.13) for Shannon, 0.00 (-0.20 to 0.20) for Simpson and -0.56 (-1.12 to 0.01) for phylogenetic

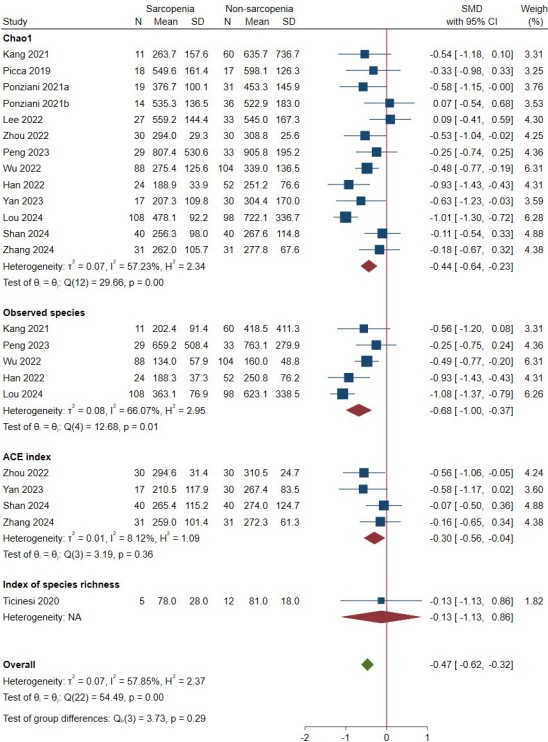

**Fig 2. Forest plots of richness in the gut microbiota of individuals with sarcopenia compared with those who were not.**

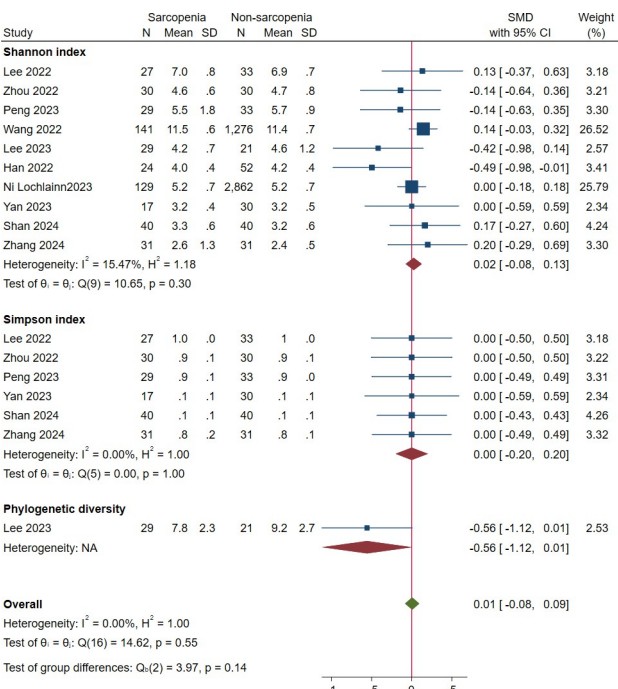

**Fig 3. Forest plots of diversity in the gut microbiota of individuals with sarcopenia compared with those who were not.**

diversity. No significant differences were observed between these indices (*P* = 0.14). The sensitivity analysis showed that results were similar with a random-effect model applied, suggesting robustness of analysis to the fixed-effect model.

Subgroup analyses were performed based on region, definition of sarcopenia, research setting, age, sex, BMI, method to measure gut microbiota, publication year, and sample size. Each variable did not have a significant association with richness (S5 Table), or diversity (S6 Table).

## Beta diversity

A total of 13 studies involving 2,978 participants were included in this part (Table 2) [20, 25, 27–29, 31–33, 36–40]. Differences in the beta diversity between sarcopenia and non-sarcopenia groups were observed in 84.6% of studies (11 of 13) and 97.3% of participants (2,899 of 2,978). In addition, although Peng *et al.* [40] found no difference between HF patients with and without sarcopenia, they found a significant difference between patients with HF plus sarcopenia and healthy participants. Taken together, these results suggest there is reliable evidence for difference between sarcopenia and non-sarcopenia in terms of beta diversity.

## Differentially abundant microbial taxa

Thirteen studies, 3,498 participants, compared the relative abundance of gut microbes between sarcopenia and non-sarcopenia [25–28, 31–35, 37–41]. Of these, 8 taxa were differently abundant at phylum level in 11 studies [26, 27, 29, 30, 33, 34, 36–40]; 30 at family level in 10 studies [26, 29, 30, 33, 34, 37–41]; 56 at genus level in 15 studies [26–29, 31–41]; and 31 at species level in 4 studies [25, 27–30, 39] (S2 Fig).

After removing microbial taxa reported only by a single study, the differences spanned 7 phyla, 23 families, 34 genera, and 3 species (Fig 4). Sarcopenia group had a lower relative

**Table 2. Methodology and findings of the included studies assessing beta diversity for the patient vs. control group comparison.**

| study | Metric | analysis | finding |
|---|---|---|---|
| Kang 2021 | Unweighted-Unifrac | PLS-DA<br>PCoA | sig. different |
| Ticinesi 2020 | Bray-Curtis | PCoA<br>PERMANOVA<br>ANOSIM | not sig. different |
| Ponziani 2021 | Weighted Unifrac | PcoA<br>PERMANOVA | sig. different |
| Lee 2022 | Bray-Curtis | NMDS | sig. different |
| Zhou2022 | Unifrac distance | PCoA<br>NMDS | sig. different |
| Han 2022 | Bray-Curtis | PcoA<br>ADONIS | sig. different |
| Peng 2023 | Bray-Curtis | PCoA<br>NMDS<br>ADONIS | not sig. different |
| Wang 2022 | Bray-Curtis | PERMANOVA | sig. different |
| Yang 2023 | Bray-Curtis | PCoA | sig. different |
| Lee 2023 | Un-weighted UniFrac distances | PcoA<br>PERMANOVA | sig. different |
| Yan 2023 | NR | PCoA | sig. different |
| Shan 2024 | UniFrac distance | PCoA | sig. different |
| Zhang 2024 | Bray-Curtis | PCoA<br>ADONIS<br>ANOSIM | sig. different |

abundance of *Prevotellaceae* at family level, as well as *Prevotella*, and *Megamonas* at genus level than the non-sarcopenia group. In addition, *Acidaminococcus*, *Lawsonibacter*, *Alistipes*, and *Prevotella copri* were altered in a consistent direction in only 2 studies, which require further investigation.

## Discussion

To our knowledge, this is the first meta-analysis of clinical studies to investigate GM alterations in sarcopenia with the aim of finding specific bacteria which may serve as biomarkers. With this study, sarcopenia was shown to be correlated with altered structure of GM, as well as a moderate reduction of species richness. The detailed analysis of the gut microbial differential abundance identified a loss of *Prevotellaceae*, *Prevotella*, and *Megamonas* in individuals with sarcopenia compared with those who were not.

With regard to beta diversity, two of studies included in our meta-analysis were not significantly different [25, 40]. Of note, the finding reported by Ticinesi et al. [25] may not be robust because of the small sample size (only 5 patients in the sarcopenia group were pooled), which limited this study's ability to assess between-individual diversity adequately. For the study conducted by Peng *et al.*, [40] although there was no difference between HF patients with and without sarcopenia, the beta diversity was reduced both in them as compared to the control group. In fact, previous studies have demonstrated that the altered composition of GM was also associated with HF [42, 43], which may overlap with alterations caused by sarcopenia, thus masking the connection between sarcopenia and GM diversity. Overall, beta diversity was observed in 84.6% of studies and 97.3% of participants in our review. According to a series of animal models, the pathways through which GM regulated muscle mass and function may

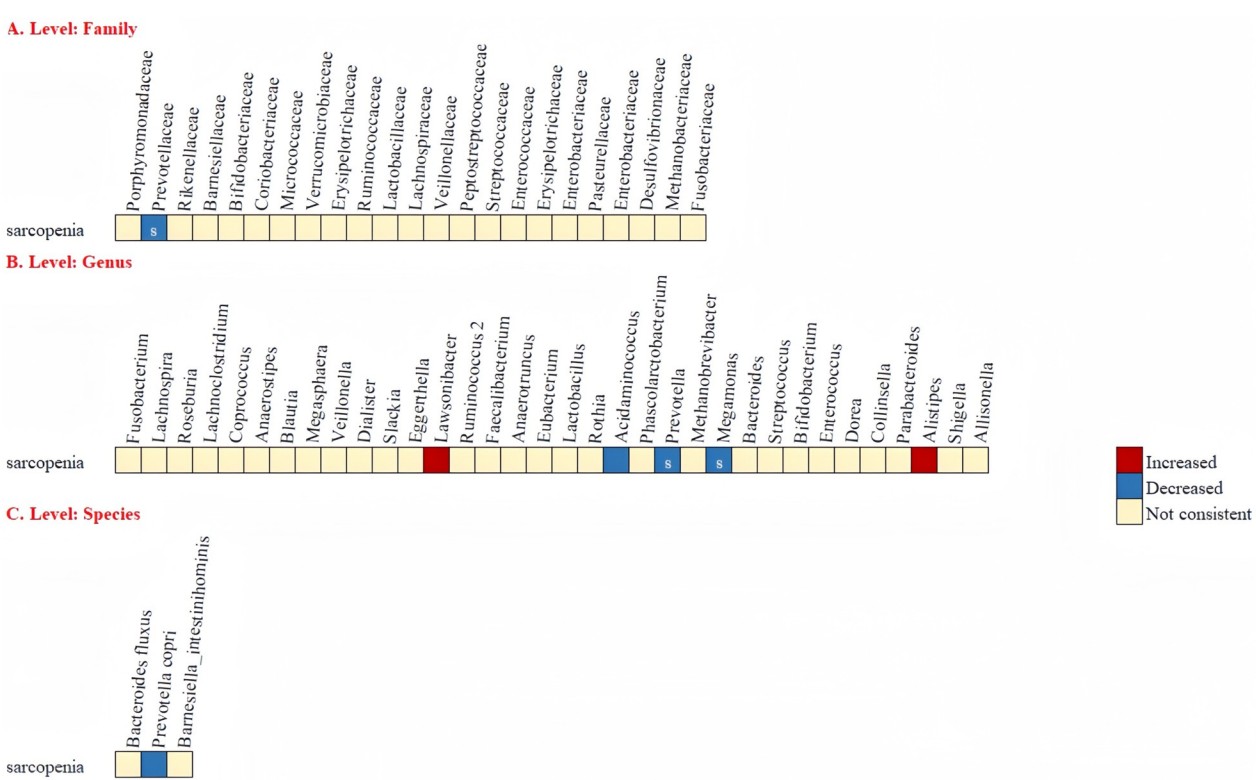

**Fig 4. Changes in relative abundance of microbial taxa reported by at least 2 studies.** 'S' refers to the bacteria which have a potential relationship with sarcopenia.

include protein anabolism, mitochondrial dysfunction, chronic inflammation and immunity, SCFA production, and harmful bacterial metabolites [12].

Our study found that the relative abundance of *Megamonas* was reduced in patients with sarcopenia. The possible mechanisms for this relationship were as follows: 1) the decrease in *Megamonas* means the reduced expression of genes that produce a high level of the SCFAs which have shown a strong correlation with muscle mass. For example, SCFAs could promote muscle protein synthesis via the mammalian target of rapamycin (mTOR)/insulin-like growth factor-1 (IGF-1) pathway [15]; SCFAs could enhance muscle differentiation and reduce muscle atrophy through the inhibition of histone deacetylase [44]; SCFAs could contribute to the improvement of muscle endurance by increasing fatty acid oxidation [45]; and SCFAs could also increase muscle glycogen levels [11]. 2) *Megamonas* was associated with energy metabolism. Several studies have shown that *Megamonas* was over-represented in overweight and obese respondents, regardless of children and adults [46–48]. The loss of *Megamonas* may result in low BMI which is an important risk factor for sarcopenia [49]. On the basis of these mechanisms, the decreased *Megamonas* was also observed in frailty—a geriatric syndrome similar to sarcopenia [50, 51], which provided an additional supplement to the validity of our results. Of note, *Megamonas* has not previously been reported as a dominant genus in European and American populations, but was found in Chinese and Japanese populations [52, 53]. Considering that our positive result was pooled through Chinese studies conducted by Kang *et al.* [33], Zhou *et al.* [39], and Peng *et al.* [40], the loss of *Megamonas* in sarcopenia may only be a characteristic of Asian populations.

Another unexpected finding of this study was that family-level *Prevotellaceae*, genus-level *Prevotella* as well as species-level *Prevotella copri* were all significantly lower in the sarcopenia group. Currently, existing evidence suggests that the impact of *Prevotella* on sarcopenia may have an opposite mechanism. On the one hand, *Prevotella* has been linked with inflammatory conditions [54, 55], which is an important ground for the development of sarcopenia. On the other hand, in line with *Megamonas*, *Prevotella* could produce a high level of SCFAs which were involved in the change of muscle biology. On the basis of our finding, the protective effect of *Prevotella* on sarcopenia may outweigh its harmful effect. Several epidemiology studies further supported this from other aspects, in which *Prevotellaceae* was enriched in non-frail people and those with high physical functioning [56], particularly young professional athletes [57]. *Prevotella* consisting of lots of species predominates in the oral cavity except for *Prevotella copri* which is generally the more abundant in the gut. Therefore, the consistent results may be attributed to the reduced *Prevotella copri*, highlighting its important role in the development of sarcopenia.

There are several limitations in our study: 1) 73.7% of studies included in our systematic review and meta-analysis were conducted in Asian population. Due to differences in factors such as diet and race, our findings may not be applicable to western countries. 2) As studies were pooled with different patient characteristics and protocols, the heterogeneity was severe in terms of Chao 1 ($I^2$ = 57.23%) and observed species ($I^2$ = 66.07%). In order to minimize the impact of high heterogeneity, we used the random-effects model to estimate the results more conservatively and objectively. Also, we performed subgroup analyses to reduce the heterogeneity, and *P* values for interaction were more than 0.05, suggesting the concordant results. More importantly, the decreased microbial richness in sarcopenic patients was confirmed by using ACE index with the mild heterogeneity ($I^2$ = 8.12%). However, although we have adopted these measures, the very low certainty of evidence resulted from observational studies and severe heterogeneity still requires us to be cautious about the validity of the results. 3) As with any meta-analysis, our dataset was founded on each included study, and certain of the outcomes or the variables were only reported by very few studies, which may increase uncertainty of the results. 4) Most studies included in our review used 16S ribosomal RNA amplicon sequencing to examined the gut microbiota profile; however, this technique is less accurate than short-gun metagenomic sequencing, and cannot identify altered specific species, leading to reduced evidence intensity of outcome at species-level.

## Conclusions

Patients with sarcopenia were found to be associated with a moderate decrease in microbial richness compared with those without. Supplementing *Prevotellaceae*, *Prevotella*, and *Megamonas* may play a significant role in preventing and treating sarcopenia. Of note, most studies included in our systematic review and meta-analysis were cross-sectional; thus, RCTs are needed to further confirm this relationship in the future.

## Supporting information

**S1 Table. PRISMA 2020 for abstracts checklist.**
(DOCX)

**S2 Table. Search strategy and search results.**
(DOCX)

**S3 Table. The joanna briggs institute (JBI) critical appraisal checklist for analytical cross-sectional studies for assessing the quality of comparative studies in the meta-analysis.**
(DOCX)

**S4 Table. GRADE evidence profile.**
(DOCX)

**S5 Table. Subgroup analysis of comparison between sarcopenia and non-sarcopenia for richness in the gut microbiota.**
(DOCX)

**S6 Table. Subgroup analysis of comparison between sarcopenia and non-sarcopenia for diversity in the gut microbiota.**
(DOCX)

**S1 Fig. Funnel plots assessing publication bias for richness index.**
(PNG)

**S2 Fig. The relative abundance of gut microbes between sarcopenia and non-sarcopenia in each study.** A) Level: phylum; B) Level: family; C) Level: genus; D) Level: species.
(PDF)

## Author Contributions

**Conceptualization:** Kangkang Chen.

**Data curation:** Xiaohong Mai.

**Formal analysis:** Shuyi Yang.

**Methodology:** Xiaohong Mai, Shuyi Yang, Qifeng Chen.

**Project administration:** Kangkang Chen.

**Writing – original draft:** Xiaohong Mai.

**Writing – review & editing:** Xiaohong Mai.

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
