## [Decision Letter · Decision Letter 0]

25 Jun 2024

PONE-D-24-16781Gut microbial composition is altered in sarcopenia: A systematic review and meta-analysis of clinical studiesPLOS ONE

Dear Dr. Chen,

Thank you for submitting your manuscript to PLOS ONE. After careful consideration, we feel that it has merit but does not fully meet PLOS ONE’s publication criteria as it currently stands. Therefore, we invite you to submit a revised version of the manuscript that addresses the points raised during the review process.

We look forward to receiving your revised manuscript.

Kind regards,

Masoud Rahmati

Academic Editor

PLOS ONE

Journal Requirements:

2. In the online submission form, you indicated that "The datasets generated during and/or analyzed during the current study are available from the corresponding author on reasonable request."

Reviewers' comments:

Reviewer's Responses to Questions

**Comments to the Author**

1. Is the manuscript technically sound, and do the data support the conclusions?

Reviewer #1: Partly

Reviewer #2: Yes

2. Has the statistical analysis been performed appropriately and rigorously? 

Reviewer #1: Yes

Reviewer #2: Yes

3. Have the authors made all data underlying the findings in their manuscript fully available?

Reviewer #1: Yes

Reviewer #2: Yes

4. Is the manuscript presented in an intelligible fashion and written in standard English?

Reviewer #1: Yes

Reviewer #2: Yes

5. Review Comments to the Author

Reviewer #1: Dear Editor

This is a good manuscript reviewing “Gut microbial composition is altered in sarcopenia: A systematic review and metaanalysis of clinical studies”. The subject of the manuscript is fully consistent with the aims and scope of the journal « PLOS ONE». The research methodology is fully consistent with the aims declared by the authors. Their conclusions are also consistent with the set goals, however, some issues need to be reconsidered:

- Please explain all abbreviations in the abstract and manuscript.

Abstracts

- Abstract should be informative, did they have any language and publication preference?

- Keywords: are these keywords are Mesh terms? Word that serves as a keyword, as to the meaning of that condition must be a Mesh term

Introduction

-The authors suggest alpha diversity and they did not explain it, because it need to be explained in the context of gut microbiota; Alpha diversity is crucial in gut microbiota studies because it provides insights into the health and stability of the gut ecosystem. A diverse gut microbiome is often associated with better health outcomes, resilience to infections, and overall well-being, while low diversity can be linked to various diseases and conditions, such as inflammatory bowel disease, obesity, and metabolic disorders.

Methods

- What about other source of potential included articles such as grey literatures? Did the authors searched this references?

Results:

- The figure 1 (flowchart of literature search and review) should be presented as the most updated version of 2020

- Table 1 is not informative; it could summarize included studies data.

Discussion

Authors should also acknowledge some serious limitations of the study:

1. the vast majority of studies originated from Eestern countries; thus, extrapolation of these results to Wastern populations is questionable.

2. significant heterogeneity was encountered perhaps due to various regimens, doses, duration, center settings, populations enrolled etc. calling for cautious interpretation of the results. This is a serious limitation and should be included because it may significantly undermine the validity of results.

3. many of the studies suffer from significant sources of bias and this should be also taken into consideration

4. the effect in many occasions was assessed by very few studies; thus, the evidence to support it is low and should be mentioned.

Authors should also include an Appendix section according to the full PRISMA Guidelines checklist, where respective sections of the manuscript are presented.

-The quality of evidence should be evaluated following the GRADE approach.

Reviewer #2: Dear writers

The gut-muscle axis theory has attracted the attention of researchers in recent years, and conducting a meta-analysis study is necessary for general conclusions. Thank you for choosing this topic.

The manuscript is well written, but revise these points to improve it.

1. It is better to focus on the gut-muscle axis in the introduction of the abstract, instead of reporting the problem of increasing the number of studies in recent years. This is a more attractive start.

2. The abstract of each article should express the study. Your conclusion is very general and it is better to be in the context of the investigated variables in the study and related to the subject.

2. The relationship between intestinal microbiota and muscles, as well as its process in sarcopenia, is not well explained in the introduction. The necessity of this study should be properly reported.

3. The results are reported correctly, but the discussion and conclusions are weak and you could have pointed to more communication mechanisms.

6. PLOS authors have the option to publish the peer review history of their article (what does this mean?). If published, this will include your full peer review and any attached files.

Reviewer #1: No

Reviewer #2: No

---

## [Author Response · Author response to Decision Letter 0]

10 Jul 2024

Dear editors and reviewers:

Thank you for your comments concerning our manuscript entitled “Gut microbial composition is altered in sarcopenia: A systematic review and meta-analysis of clinical studies” (ID: PONE-D-24-16781). Those comments are all valuable and very helpful for revising and improving our paper, as well as the important guiding significance to our researches. We have studied comments carefully and have made correction which we hope meet with approval. The main corrections in the paper and the responds to your comments are as following:

Journal Requirements:

Response: According to the journal requirements, we have modified our manuscript’s style.

2.In the online submission form, you indicated that "The datasets generated during and/or analyzed during the current study are available from the corresponding author on reasonable request."

Response: Other researchers can find all data underlying the findings described in our manuscript within our manuscript itself.

3.PLOS requires an ORCID iD for the corresponding author in Editorial Manager on papers submitted after December 6th, 2016. Please ensure that you have an ORCID iD and that it is validated in Editorial Manager. To do this, go to ‘Update my Information’ (in the upper left-hand corner of the main menu), and click on the Fetch/Validate link next to the ORCID field. This will take you to the ORCID site and allow you to create a new iD or authenticate a pre-existing iD in Editorial Manager. Please see the following video for instructions on linking an ORCID iD to your Editorial Manager account: https://www.youtube.com/watch?v=_xcclfuvtxQ

Response: The corresponding author’s account has been linked to an ORCID iD.

4.We note that you have included the phrase “data not shown” in your manuscript. Unfortunately, this does not meet our data sharing requirements. PLOS does not permit references to inaccessible data. We require that authors provide all relevant data within the paper, Supporting Information files, or in an acceptable, public repository. Please add a citation to support this phrase or upload the data that corresponds with these findings to a stable repository (such as Figshare or Dryad) and provide and URLs, DOIs, or accession numbers that may be used to access these data. Or, if the data are not a core part of the research being presented in your study, we ask that you remove the phrase that refers to these data.

Response: We have removed the phrase “data not shown” since the data are not a core part of the research.

5.Please include captions for your Supporting Information files at the end of your manuscript, and update any in-text citations to match accordingly. Please see our Supporting Information guidelines for more information: http://journals.plos.org/plosone/s/supporting-information.

Response: According to the Supporting Information guidelines, we have listed the supporting information captions at the end of the manuscript in a section titled “Supporting information”.

Reviewer 1

- Please explain all abbreviations in the abstract and manuscript.

Response: Thanks to the reviewer for this comment. After inspecting our manuscript, we have completed some abbreviations without full name (please see line 210-213, 294). Also, we have added explanations of all abbreviations on the title page (please see page 1). 

Abstracts

- Abstract should be informative, did they have any language and publication preference?

Response: Thanks to the reviewer for this comment. In order to make our abstract more informative and specific, we have modified the background, the method and the conclusion of the abstract according to the PRISMA 2020 statement. Please see line 35-65. 

- Keywords: are these keywords are Mesh terms? Word that serves as a keyword, as to the meaning of that condition must be a Mesh term

Response: According to the reviewer’s suggestion, we used Mesh terms as the keywords. Please see line 66.

Introduction

-The authors suggest alpha diversity and they did not explain it, because it need to be explained in the context of gut microbiota; Alpha diversity is crucial in gut microbiota studies because it provides insights into the health and stability of the gut ecosystem. A diverse gut microbiome is often associated with better health outcomes, resilience to infections, and overall well-being, while low diversity can be linked to various diseases and conditions, such as inflammatory bowel disease, obesity, and metabolic disorders.

Response: Thanks to the reviewer for this comment. Based on your and another reviewer’s suggestions, we have revised the introduction section. Due to limited space and coherence of content, we put the introduction of alpha diversity in the third paragraph, please see line 96-98. 

Methods

- What about other source of potential included articles such as grey literatures? Did the authors searched this references?

Response: Thanks to the reviewer for this comment. Currently, we search PubMed using a retrieval formula “((grey literature[Text Word]) AND (meta[Text Word]))” and we find 3,648 records, only accounting for 0.98% (3,648/371,868) of all meta-analyses when using “(meta[Text Word])”. Furthermore, according to a paper published by Hartling et al.,[1] they found that including non-English reports, unpublished studies and dissertations had little impact on results of meta-analyses. Given that there was limited evidence on the contribution of grey literature, we did not include them in our study. 

Results:

- The figure 1 (flowchart of literature search and review) should be presented as the most updated version of 2020

Response: According to the reviewer’s suggestion, we have modified the figure 1 using the most updated PRISMA Flow Diagram 2020.

- Table 1 is not informative; it could summarize included studies data.

Response: According to the reviewer’s suggestion, we have added the studies data into the Table 1. 

Discussion

Authors should also acknowledge some serious limitations of the study:

1. the vast majority of studies originated from Eestern countries; thus, extrapolation of these results to Wastern populations is questionable.

Response: According to the reviewer’s suggestion, we have added this limitation to the discussion section. Please see line 342-345.

2. significant heterogeneity was encountered perhaps due to various regimens, doses, duration, center settings, populations enrolled etc. calling for cautious interpretation of the results. This is a serious limitation and should be included because it may significantly undermine the validity of results.

Response: According to the reviewer’s suggestion, we have added this limitation to the discussion section. Please see line 345-355.

3. many of the studies suffer from significant sources of bias and this should be also taken into consideration.

Response: Thanks to the reviewer for this comment. As studies included in our systematic review and meta-analysis were cross-sectional, the Joanna Briggs Institute (JBI) critical appraisal checklist for analytical cross-sectional studies was used to assess the methodological quality of included studies. According to the JBI critical appraisal checklist, almost all studies themselves were high in terms of the methodological quality. For differences in patient characteristics and research designs between studies, we have added this limitation to my manuscript on the basis of your previous comment (please see line 345-355).

4. the effect in many occasions was assessed by very few studies; thus, the evidence to support it is low and should be mentioned.

Response: According to the reviewer’s suggestion, we have added this limitation to the discussion section. Please see line 355-357.

Authors should also include an Appendix section according to the full PRISMA Guidelines checklist, where respective sections of the manuscript are presented.

Response: According to the reviewer’s suggestion, we have submitted the PRISMA2020 checklist. Please see S1 Table (line 110-111) in the supplemental file.

-The quality of evidence should be evaluated following the GRADE approach.

Response: Thanks to the reviewer for this comment. We have evaluated the certainty of the body of evidence associated with alpha diversity. Please see the Materials and methods section (line 149-158), the Results section (line 226-228), and the Discussion section (line 352-355).

Reviewer 2

1. It is better to focus on the gut-muscle axis in the introduction of the abstract, instead of reporting the problem of increasing the number of studies in recent years. This is a more attractive start.

Response: Thanks to the reviewer for this comment. We have modified the background of the abstract. Please see Line 36-41.

2. The abstract of each article should express the study. Your conclusion is very general and it is better to be in the context of the investigated variables in the study and related to the subject.

Response: According to the reviewer’s suggestion, we have modified the conclusion of the abstract. Please see Line 61-63.

3. The relationship between intestinal microbiota and muscles, as well as its process in sarcopenia, is not well explained in the introduction. The necessity of this study should be properly reported.

Response: According to the reviewer’s suggestion, we have re-written the introduction. Please see the red font in the introduction.

4. The results are reported correctly, but the discussion and conclusions are weak and you could have pointed to more communication mechanisms.

Response: According to the reviewer’s suggestion, we have re-written the discussion and the conclusion. Please see the red font in both sections.

We tried our best to improve the manuscript and made some changes in the manuscript. We appreciate for Reviewers’ warm work earnestly and hope that the correction will meet with approval. Once again, thank you very much for your comments and suggestions.

---

## [Decision Letter · Decision Letter 1]

23 Jul 2024

Gut microbial composition is altered in sarcopenia: A systematic review and meta-analysis of clinical studies

PONE-D-24-16781R1

Dear Dr. Chen,

We’re pleased to inform you that your manuscript has been judged scientifically suitable for publication and will be formally accepted for publication once it meets all outstanding technical requirements.

Kind regards,

Masoud Rahmati

Academic Editor

PLOS ONE

Additional Editor Comments (optional):

Reviewers' comments:

Reviewer's Responses to Questions

**Comments to the Author**

1. If the authors have adequately addressed your comments raised in a previous round of review and you feel that this manuscript is now acceptable for publication, you may indicate that here to bypass the “Comments to the Author” section, enter your conflict of interest statement in the “Confidential to Editor” section, and submit your "Accept" recommendation.

Reviewer #2: All comments have been addressed

2. Is the manuscript technically sound, and do the data support the conclusions?

Reviewer #2: Yes

3. Has the statistical analysis been performed appropriately and rigorously? 

Reviewer #2: Yes

4. Have the authors made all data underlying the findings in their manuscript fully available?

Reviewer #2: Yes

5. Is the manuscript presented in an intelligible fashion and written in standard English?

Reviewer #2: Yes

6. Review Comments to the Author

Reviewer #2: (No Response)

7. PLOS authors have the option to publish the peer review history of their article (what does this mean?). If published, this will include your full peer review and any attached files.

Reviewer #2: No

---

## [Editor Report · Acceptance letter]

25 Jul 2024

PONE-D-24-16781R1 

PLOS ONE

Dear Dr. Chen, 

I'm pleased to inform you that your manuscript has been deemed suitable for publication in PLOS ONE. Congratulations! Your manuscript is now being handed over to our production team.

Kind regards, 

on behalf of

Dr. Masoud Rahmati 

Academic Editor

PLOS ONE